# Clonality, Mutation and Kaposi Sarcoma: A Systematic Review

**DOI:** 10.3390/cancers14051201

**Published:** 2022-02-25

**Authors:** Blanca Iciar Indave Ruiz, Subasri Armon, Reiko Watanabe, Lesley Uttley, Valerie A. White, Alexander J. Lazar, Ian A. Cree

**Affiliations:** 1International Agency for Research on Cancer (IARC), World Health Organization, 69372 Lyon, France; sarmon2003@gmail.com (S.A.); watanabe0153@yahoo.co.jp (R.W.); valerieawhite@gmail.com (V.A.W.); creei@iarc.fr (I.A.C.); 2School of Health and Related Research (ScHARR), University of Sheffield, Sheffield S1 4DA, UK; l.uttley@sheffield.ac.uk; 3Department of Pathology, The University of Texas MD Anderson Cancer Center, Houston, TX 77030, USA; alazar@mdanderson.org

**Keywords:** Kaposi, sarcoma, clonality, reactive, DNA, HHV8

## Abstract

**Simple Summary:**

Kaposi’s sarcoma (KS) is a rare tumour of uncertain nature. It may be a true cancer or an aggressive viral lesion, which can regress in many patients, and there is a real need for more research on the subject. This systematic review aimed to summarize the available evidence on somatic mutations and clonality within KS to assess whether KS is a neoplasm or not, concluding that knowledge is currently insufficient to determine whether KS is a clonal neoplasm (sarcoma), or simply an aggressive reactive virus-driven lesion.

**Abstract:**

Background: It remains uncertain whether Kaposi sarcoma (KS) is a true neoplasm, in that it regresses after removal of the stimulus to growth (as HHV8) when immunosuppression is reduced. We aimed to summarize the available evidence on somatic mutations and clonality within KS to assess whether KS is a neoplasm or not. Methods: Medline and Web of Science were searched until September 2020 for articles on clonality or mutation in KS. Search strings were supervised by expert librarians, and two researchers independently performed study selection and data extraction. An adapted version of the QUADAS2 tool was used for methodological quality appraisal. Results: Of 3077 identified records, 20 publications reported on relevant outcomes and were eligible for qualitative synthesis. Five studies reported on clonality, 10 studies reported on various mutations, and 5 studies reported on chromosomal aberrations in KS. All studies were descriptive and were judged to have a high risk of bias. There was considerable heterogeneity of results with respect to clonality, mutation and cytogenetic abnormalities as well as in terms of types of lesions and patient characteristics. Conclusions: While KS certainly produces tumours, the knowledge is currently insufficient to determine whether KS is a clonal neoplasm (sarcoma), or simply an aggressive reactive virus-driven lesion.

## 1. Introduction

Kaposi sarcoma (KS) is an unusual neoplasm, which presents with clinical manifestations that range from those that are slowly progressive and confined to the skin, to an aggressive tumour that invades visceral organs. The clinicopathological characteristics are usually described in terms of the location of the lesions (lymph node, visceral or skin), and the clinical stage of the disease (patch, plaque or nodular). Epidemiologically, KS presents in several forms: (a) a classic form predominantly in older men; (b) an endemic form in young men and children from central Africa; (c) an iatrogenic form secondary to treatment with immunosuppressive drugs, such as steroids; and (d) an epidemic, HIV/AIDS-associated form [1].

KS was first described at the end of the 19th century as mainly occurring in elderly males from Mediterranean regions but was later reported in East and Central Africa with presentation in younger people [1,2]. It was a rare tumour in certain countries, such as North America and Europe, before the rise of HIV/AIDS in the 1980s, when its incidence increased dramatically and this tumour became characteristic as being an AIDS-related malignancy [1,3]. Currently, while its incidence varies greatly in different regions of the world, KS remains a rare disease. Most of the cases seen in Europe, North America and Africa are young children, renal allograft recipients, patients on immunosuppressive therapy and HIV-infected patients [4]. 

KS became the most common HIV-related “malignancy” worldwide, although HIV alone could not explain all of the cases. Its occurrence in patients on long-term immunosuppressive therapy and the predominance in males and HIV-negative immunocompetent individuals with increased risk for Sexual Transmitted Diseases (STDs) suggested the implication of another transmissible agent. Eventually human herpesvirus-8 (HHV8) was identified as the virus associated with KS in all different groups, including individuals in Eastern Europe, Africa and the United States [5].

Depending on the clinical subtype, age-standardized incidence rates ranged from 0.3 in Eastern Asia to 8.5 ASR per 100,000 in Eastern Africa in 2020 [6]. Various African regions reported an important increase in the incidence of KS over the last decades [7,8], and western countries also described new patterns with more cases in young and older males [4,6]. There are several well-known risk factors that include Human Herpes Virus 8 (HHV8) infection, immune deficiencies (including those caused by treatments), environmental factors related to skin hygiene and/or skin disease and diabetes [1,4]. Improved control of certain risk factors, such as antiviral therapy for HIV infection, has helped control the number of cases of some epidemiological forms in the last two decades.

A neoplasm is usually defined as “an abnormal mass of tissue, the growth of which exceeds and is uncoordinated with that of the normal tissues, and persists in the same excessive manner after cessation of the stimulus which evoked the change” [9]. KS does not meet the second part of this definition, since it does not persist after the inciting virus, HHV8 [10], has been eliminated. This is the usual clinical result of successful anti-HIV treatment in patients with KS and considering the definition, could be taken to mean that KS is not a neoplasm.

Intersecting with this established definition is the concept of clonality, which is not part of the long-accepted definition of a neoplasm and states that cancers evolve by an iterative process of clonal expansion [11]. The dogma that all neoplasms must be monoclonal is widely accepted and used for diagnosis as well as cancer research. Thus, if KS is in fact an oligoclonal proliferation of cells and continues to be accepted as such, then unless the importance of clonality within neoplasms is reassessed, this has consequences for its classification as a sarcoma. Understanding of the clonal origins of tumours is critical not only for a correct classification but also for the development of effective strategies to diagnose, treat and prevent cancer [12].

There are other tumours with similar issues. For example, transient neoplasia is an increasingly used concept, where diseases, such as nodular fasciitis, have a translocation but still spontaneously resolve in many cases [13]. This is also the case for keratoacanthoma, a tumour that is histologically indistinguishable from well-differentiated squamous cell carcinoma and is diagnosed on the basis of length of history but has the ability to resolve spontaneously. 

It is likely to be viral in origin; however, there is still a debate about which virus is involved: HPV is the main suspect as in KS. There are other cellular proliferations that are oligoclonal and treated seriously but not regarded as cancers. Infectious mononucleosis [14] is driven by Epstein–Barr virus (EBV) but is certainly not regarded as leukaemia—and it resolves as the immune response to EBV leads to a reduction in viral load [15].

We know that most cancers start as a clonal proliferation from a single cell and acquire mutations with growth/progression, something that was described early by Knudson in his studies of retinoblastoma [16]. However recent research is broadening our knowledge, and many studies suggest that cancer develops as a result of somatic mutation and clonal selection [11,17,18]. The high frequency of cancer-driving mutations in normal tissues sometimes appears to indicate that somatic mutation and clonal selection alone are insufficient to explain cancer development and that other factors must be required to promote carcinogenesis.

KS is currently classified as a neoplasm by the WHO Classification of Tumours (WCT); however, this constitutes a controversial topic that continues to be reviewed periodically. Any future decision on its status will be informed by the best available evidence. The aim of this systematic review is to summarize any available published scientific evidence on KS with regard to somatic mutations and clonality in order to determine whether KS can be defined as a neoplasm or not. The results of this evidence synthesis will serve to inform future decisions of the WCT and detect possible evidence gaps.

## 2. Materials and Methods

A systematic review was conducted to identify peer-reviewed articles of data on clonality or somatic mutations in KS following a protocol registered with PROSPERO (CRD42018087595). This review was performed following the PRISMA (preferred reporting items for systematic reviews and meta-analyses) guidelines and considering the recently published update of these guidelines in 2020 [19].

### 2.1. Literature Search and Study Selection

Tailored conceptual strings of relevant keywords and database-specific terms were devised for the major concepts of Kaposi Sarcoma, Clonality and mutations. Conceptual strings were combined with Boolean operators using appropriate MeSH headings and filters to search the Medline and Web of Science databases for peer-reviewed articles published up to 01 September 2020, without date restrictions. The Cochrane library was also consulted, and the reference lists of relevant articles were checked for additional studies. The concepts and keywords used to inform the search strategy are detailed in Table 1, and the full search strategy is presented in Appendix A.

Studies were excluded if they exclusively focused on: (1) Kaposi Sarcoma Herpes Virus (KSHV) mutations or viral clonality; (2) on cell lines/cultures; (3) describing tumour phenotype, proteins, receptors, vascular markers etc.; (4) mitochondrial genomes; or (5) the epidemiology or aetiology of KS; or if they were not original research, such as editorials, letters, narrative reviews and book chapters. Titles and abstracts were screened by two reviewers independently (I.C. and B.I.I.), and the full text of potentially eligible articles was assessed to decide final inclusion by both of them. The lists of included articles were compared and differences about the selected studies were resolved by consensus.

### 2.2. Data Extraction

All data were extracted by two researchers independently (B.I.I. and A.L.) into standardized data extraction forms. Discrepancies in data extraction were resolved by discussion and consensus. Relevant information was recorded from the selected publications, including the author(s), publication year, country where the study was performed, baseline population characteristics and demographics, study methodology, methods for clonality or mutation testing, main results on clonality, somatic mutations or chromosomal aberrations (only descriptive outcomes due to the nature of the studies) as well as COI disclosure. We registered frequency measures and ordered the rest of outcomes strategically in order to facilitate interpretation of the results. Data were recorded and compiled using Microsoft Excel.

### 2.3. Assessment of Risk of Bias of Included Studies

Risk of bias of included individual studies was performed using an ad hoc adapted version of the QUADAS2 tool for the Quality Assessment of Diagnostic Accuracy Studies [20]. This assessment included a general appraisal of the external and internal validity of the selected studies, as well as of the biases relevant to studies focusing on diagnostic determinations, adapted ad hoc to the retrieved study designs. One reviewer (B.I.I.) scored items as low, high, or unclear risk of bias; the results were discussed with the review team and resolved by consensus. The final evaluation of all studies was included into the summary of findings tables after resolving differences through arbitration of a third reviewer (I.A.C.).

### 2.4. Synthesis of Results

The results were tabulated in Excel sheets and summary of findings tables were drafted to present the main outcomes for each of the included studies. Outcomes were grouped based on their focus: clonality, somatic mutations or chromosomal aberrations. A descriptive analysis was performed for a qualitative synthesis of the results. The performance of a meta-analysis to calculate an adjusted pooled estimate was ruled out because the descriptive nature of the retrieved studies means that a pooled summary of the data would not be useful. Additional important methodological heterogeneity between the studies from the reported outcome measures rendered formal quantitative synthesis inappropriate. Instead, the aim was to narratively assess and combine the studies in order to derive clinically meaningful conclusions about the nature of KS.

## 3. Results

Of the 3077 records identified and screened through the database searches, 25 publications were eventually deemed as relevant and retrieved for full text inspection and qualitative synthesis (Figure 1). Five of these records were conference abstracts and were therefore excluded for not being peer-reviewed research.

Results of the included studies are summarised in Table 2, Table 3 and Table 4 according to the following outcome categories: (case series) reporting outcomes on clonality, reporting on mutations and reporting on chromosomal aberrations. After applying the eligibility criteria, 5 studies reporting on clonality, 10 studies on various mutations and 5 studies reporting on chromosomal aberrations in KS were selected. The majority of publications were traditional case series; however, two of the selected studies aimed applied a different study design. 

Tornesello et al. [21] performed a case control study investigating TP53 mutations in classic, epidemic and endemic KS cases, and Cerimele et al. [22] applied a prospective design to a case series obtained from a large cohort study in Sardinia. Nevertheless, only descriptive analysis of the genetic factors was provided by these studies as with the other case series, since the main aim of the studies relied on investigating the epidemiology of endemic cases. Studies provided descriptive results with frequency measures, and only a few studies provided analysis of the differences between groups.

The studies were performed using data from 13 different countries, and eight were international collaborations (Table 2, Table 3 and Table 4). The first study was published in 1984, and the most recent one was from 2015. Together, the studies enrolled 498 cases of KS and 421 controls; 46 cases and 12 controls to investigate clonality, 238 cases and 408 controls to study mutations and 214 cases and 1 control for chromosomal aberration studies. Studies on clonality included only female cases and were of poor reporting quality; often not reporting basic demographic variables, such as age. Additionally, the studies on mutations and chromosomal aberrations often did not provide adequate patient population and sample descriptions, making it difficult to assess the representativeness of the samples (Summary of Findings Table 2, Table 3 and Table 4).

**Table 2 cancers-14-01201-t002:** Summary of findings: studies reporting on clonality.

Included Studies (All Case Series) Reporting Outcomes on Clonality					
	Study 1	Study 2	Study 3	Study 4	Study 5
Authors	Delabesse et al. [23]	Ding et al. [24]	Gil et al. [25]	Rabkin et al. [26]	Rabkin et al. [27]
Year (Country)	1997 (France)	2015 (China)	1998 (USA)	1995 (USA)	1997 (Zambia and USA)
Sample Description (Ca+Ct)	7 + 6 ♀ skin biopsies Age not reported.	14 + 1 ♀ Mean age 48.4 years (range 27–71)	12 ♀, 4 had multiple biopsies. Mean age 49 years (range 27–89).	3 ♀ +number Ct not described Age not reported	10 ♀, 5 lesion samples + 1 Ct for each Median age 26 years (range 20–35).
Case Classification	Clinically: 4 Classic KS, 3 AIDS SG: 1 macular, 3 plaque, 3 nodular	Clinically: 6 Classic KS, 8 AIDS SG: 2 macular, 11 plaque, 1 nodular Other: all cases HHV-8 +	Clinically: 2 Classic KS and 10 AIDS (8 advanced disease, 3 history of glucocorticoid use) SG: not reported.	Clinically: all HIV type 1-sero+ SG: all nodular KS lesions	Clinically: all HIV positive (8 serological, 2 clinical diagnosis) SG: multiple nodular KS
Study Results					
Description of Investigation Used	Punch biopsies HP: HES MA: DNA extraction Clonality AS: PCR amplification, HUMARA	HP: Surgical tissues, standard histology MA: IHCS of the primary mcAB. Clonality AS: PCR amplification, HUMARA detection of single-nucleotide polymorphism sites in PGK	HP: Cutaneous tumour biopsies. HES. Review performed by, to DNA results, blinded pathologist. MA: DNA extraction. Clonality AS: Secondary PCR products analysed by electrophoresis + autoradiography	HP: Cutaneous biopsies, HES MA: DNA extraction. Clonality AS: X chromosome inactivation assay (HUMARA)	HP: Cutaneous biopsies, HES MA: DNA extraction. Clonality AS: X chromosome inactivation assay (HUMARA)
Main Results	Descriptive: All 7 patients were heterozygous for HUMARA Clonality: All polyclonal pattern of inactivation	Descriptive: 2 Ca failed to amplify HUMARA, 11 analysed HUMARA, 5 KS PGK Clonality: 1Ca+1Ct polyclonal pattern, rest monoclonal Ca with no significant differences between groups (*p* > 0.05)	Descriptive: 41 different regions from 24 biopsies were studied. Clonality: 5 Ca clonal, 2 Ca inactivate, 7 Ca polyclonal pattern of inactivation, 2 Ca both clonal/polyclonal inactivation	Descriptive: All 3 patients heterozygous androgen receptor Clonality: 2 of 3 Ca monoclonal pattern	Descriptive: 2 Ca excluded (homozygous HUMARA), 8 Ca (40 tumours, 32 studied) Clonality: 23 (85%) tumours had unbalanced methylation (predominance one allele), 6 Ca (23 tumours) concordant methylation (≤0.00001).
Conclusions	It is a polyclonal cell proliferation.	Suggest a clonal neoplasm.	Suggest a clonal neoplasm, but polyclonal inactivation pattern observed may be premalignant stage or false negative results.	Suggest a clonal neoplasm (at least in AIDS Ca).	Data indicate monoclonal cancer.

♀: female, AIDS: AIDS associated Kaposi Sarcoma, Ca: Cases Ct: Controls, Clonality AS: Clonality assay, HES: Haematoxylin and eosin staining, HP: Histopathology, HUMARA: HUMARA gen polymorphism analysis, IHCS: Immunohistochemical staining, KS: Kaposi Sarcoma, MA: Molecular analysis, mcAB: monoclonal antibody, PGK: Phosphoglycerate kinase gene, SG: Staging. When data not reported in the table means the original article did not provide them.

**Table 3 cancers-14-01201-t003:** Summary of findings: studies reporting on mutations.

Included Studies Reporting Outcomes on Mutations										
	Study 1	Study 2	Study 3	Study 4	Study 5	Study 6	Study 7	Study 8	Study 9	Study 10
Authors	Nicolaides et al. [28]	Kiuru-Kuhlefelt et al. [29]	Cerimele et al. [22]	Tornesello et al. [21]	Huang et al. [30]	Guttman-Yassky et al. [31]	Feller et al. [32]	Cordiali-Fei et al. [33]	Li, Jian J. [34]	Scinicariello et al. [35]
Year (Country)	1994 (USA)	2000 (Finland + USA)	1984 (Italy)	2009 (Italy + Uganda + Greece)	1993 (USA)	2012 (USA + Israel)	2014 (USA)	2014 (Italy)	1997 (USA)	1994 (USA)
Study Design	Case series	Case series	Case series	Case control	Case series	Case series	Case series	Case series	Case series	Case series
Sample Description (Ca + Ct)	31 Ca No demographics reported.	12 Ca 10 ♂, 2 ♀ Age: NR	(65) 22 + 220 49 ♂, 16 ♀ Age: range 57–82 years	67 + 150 No data disaggregated for ♂♀ Median age: 75 (29.5–47) years	38 + 10 No demographics reported.	9 + 4 6 ♂, 7 ♀ Mean age: Ca 69.5 (20–86) years	24 + 17 21 ♂, 3 ♀ Mean age: Ca 59 (26–86) years	3 Ca same family 2 ♂, 1 ♀ Age: 63, 64, 40 years	15 + 5 No demographics reported.	17 Ca No demographics reported.
Case Classification	Clinically: 24 HIV+, 7 HIV− SG: 16 nodular, 6 patch, 9 plaque	Clinically: 6 ♂ HIV+ or AIDS	Clinically: all endemic KS (Sardinia).	Clinically: 33 classic, 2 iatrogenic, 19 epidemic and 13 epidemic Other: all HHV8+	Clinically: 31 AIDS, 7 classic	Clinically: All classic SG: 4 early, 2 mixed and 3 nodular	Clinically: 6 HIV+, rest unknown SG: 9 patch, 7 plaque, 8 nodular	Clinically: All HIV− Other: all HHV-8+	Clinically: All AIDS	Clinically: 10 Classic, 7 HIV+
Study Results										
Description of Investigation Used	GenA: PCR-SSCP (Orita procedure) to detect mutations	GenA: PCR HHV-8 sequence specific primers, aCGH, digital image analysis, FISH, IHCS	GenA: THLA-ABC typing with 162 antisera. HLA-DR and MT typing using 47 antisera. StatA: significance calculated using X’ test+Wolf’s relative risk test.	HP: Cutaneous biopsies GenA: TP53 genotype at codon 72 StatA: Fisher’s exact or X’ test for comparison of Ca/Ct. Student’s *t* test for age differences	HP: Biopsy/autopsy samples. GenA: R T-PCR, DNA sequencing, ICHS, Southern blot hybridization	HP: Punch biopsies GenA: Gene chip analysis, DNA microarray analysis, IHCS, Immunoflouresence+, obtaining expression profile LEC and BEC gene signature	HP: All sections reviewed by dermatopathologists GenA: FISH, IHCS	Not reported	GenA: RT-PCR + PCR-SSCP in immunoperoxidase stains	GenA: DNA+ PCR amplification, HPV DNA detection, p53 direct sequencing with (ɣ32P) ATP end-labelled primers
Main Results	Genetics: 10 Kras overexpression (3 Kras amplification, 7 various mutations Kras exon)	Genetics: 4 recurrent again at 11q13; 4 containingFGF4 and INT2 (expression of FGF4 and INT2 was found in 9 and 3 Ca respectively).	Genetics: No differences in A, B, C antigen frequency (DR5 72.7% Ca, 23.1% Ct, *p* < 0.0001; DR3 9.1% Ca 53.6% Ct, *p* < 0.01; DR5 36.4% Ca 18.1% Ct, *p* not significant)	Genetics: African Ca: PHoZ 50%, PHtZ 31.8%, AHoZ 18.2%, (*p* = 0.1872) Caucasian Ca: PHoZ 6.7, PHtZ 55.6, AHoZ 37.8%, (*p* = 0.0567). Stratified by HIV: No significant differences in alleles	Genetics: Int-2 expressed in 21 Ca (55.2%), NASalt in 8. Most variations in int-2 cDNA located in exon 1 (four in exons 2 and 3)	Genetics: Gene expression level markers gradually increased from normal through all KS stages, particularly LEC genes.	Genetics: c-myc amplification in all Ca, IHCS positive for c-Myc in 13 Ca (54%)	Genetics: IL-6 promoter polymorphism G-174C (2 ♂ HtZ, ♀ HoZ)	Genetics: 4 Ca p53 in nuclei+cytoplasm, 5 MDM2 in the nuclei (2 IHCS+ p53, 3 IHCS− p53 protein)	Genetics: 4 Ca (23%) HPV DNA detected (1 AIDS), 5 Ca (24%) p53 HtZ (none in HPV +)
Conclusions	Suggests K-ras mutation plays a significant role in KS oncogenesis.	No evidence of HHV-8 integration to genome.	Preliminary evidence of structural chromosome rearrangement.	p53 polymorphism at codon 72 does not represent a RF for KS.	Int-2 expression may play a role in KS oncogenesis.	Suggests local expression of chemokines/growth factors (no clonal exp.) as oncogenesis.	No amplification of the c-myc gene detected.	Suggests that EBV can cause HHV-8 reactivation in predisposed Ca causing KS.	Suggest p53 may be involved in AIDS KS pathogenesis.	Indicate role of HPV to KS pathogenesis and p53 alteration to malignancy progression.

♂: Male, ♀: female, aCGH: array comparative genomic hybridization, AIDS: AIDS associated Kaposi Sarcoma, Ca: Cases Ct: Controls, FISH: Interphase fluorescence in situ hybridization, GenA: Genetic analysis, HP: Histopathology, PHoZ: Proline homozygous, PHtZ: Proline heterozygous, IHCS: Immunohistochemical staining, KS: Kaposi Sarcoma, NR: Not reported, NASalt: Nucleic acid sequence alterations, PCR-SSCP: Polymerase chain reaction-single-strand conformation polymorphism analysis, RF: Risk factor, R T-PCR: Reverse transcription-PCR, SG: Staging, and StatA: Statistical analysis. When data not reported in the table means the original article did not provide them.

**Table 4 cancers-14-01201-t004:** Summary of findings: studies reporting on chromosomal aberrations.

Included Studies (All Case Series) Reporting Chromosomal Aberrations					
	Study 1	Study 2	Study 3	Study 4	Study 5
Authors	Kaaya et al. [36]	Kaaya et al. [37]	Bisceglia et al. [38]	Pyakurel et al. [39]	Reizis et al. [40]
Year (Country)	2000 (Sweden + Tanzania)	1992 (Sweden + Tanzania)	1991 (UK + Italy)	2006 (Sweden + Germany + Tanzania)	1995 (Israel)
Sample Description (Ca + Ct)	32 Ca (12 Ca ploidy analysis) 22 ♂, 10 ♀ Age: range 8–68 years	20 Ca 17 ♂, 3 ♀ Mean age: 40 (up to 83) years	96 Ca (66 analysed, 143 biopsies) 69 ♂, 27 ♀ Mean age: 69 (10–89) years	27 + 1 Mean age: males 37.5 (23–65) years, females not reported	39 Ca Mean age: Iatrogenic Ca 68 (54–80) years
Case Classification	Clinically: 8 endemic, 24 AIDS SG: 24 nodular, 4 patch, 4 plaque Other: 12 Ploidy analysed Ca (6 endemic, 6 AIDS)	Clinically: 10 endemic, 10 AIDS SG: 17 nodular skin, and 3 generalised lymph node lesions	Clinically: 93% sporadic, 6% AIDS, 1% Hepatitis B	Clinically: 9 Endemic, 18 AIDS SG: 18 nodular, 9 patch	Clinically: 31 classic, 8 iatrogenic (steroid induced)
Study Results					
Description of Investigation Used	HP: Surgical tissues Mol.An: HHV-8 DNA PCR, IHCS, Ploidy by DNA flow cytometry, apoptotic cells (TUNEL-assay)	HP: Surgical tissues. Mol.An: IHCS, DNA measurements	HP: slides classified by histological criteria Mol.An: Mitoses count, Flow Cytometry (DNA aneuploidy ≥1 GO/Gl peak modal channel number)	HP: Surgical tissues Mol.An: Ligation-mediated PCR+DNA labelling, aCGH, FISH	HP: tissue blocks Mol.An: Flowcytometry
Main Results	Ploidy: All 12 diploid cellular DNA content, and low numbers of cells (1.6–8.9%) in S and G2 phase. Ploidy values similar normal cells in non-involved tissue of the same section. In contrast the malignant cell line (KS Y-1) showed a near triploid, aneuploid DNA content and a moderate proliferation rate (13% cells in S + G2 phase).	Ploidy: 70% cells contained DNA values ≥2.5, but not greater than 5C level. Both clinical types with euploid DNA pattern.	Ploidy: 6 lesions (5.8%) DNA aneuploid with a clustering around a DNA index of 1.5 (range 1.4–1.6). Increasing mitotic counts and S-phase plus G2-phase cells were seen with progression of the phase and pattern of disease. Nodular and spindle cell forms had the highest mitotic counts and S-phase plus G2-phase cells.	Chromosomal results: 20 (87%) Ca only recurrent aberration loss of Y chromosome One patch Ca showed in addition loss of Xq. Nodular showed recurrent copy number changes in chromosomes 16, 17, 21, X, Y, and other random changes.	Ploidy: 28 classic Ca showed a diploid pattern. Of 8 iatrogenic, 7 were aneuploid and 1 diploid.
Conclusions	Represents a diploid, probably reactive, cell proliferation, which progressively increases the expression of antiapoptotic factors (cellular and viral).	Corroborates previous suggestions that KS could represent a reactive process, rather than a clonal proliferation.	Suggest a low level of DNA aneuploidy, but flow cytometry does not solve the dilemma of whether KS is a hyperplastic or neoplastic process.	Support the view that KS (in males) develops into a clonal tumour yet initially is a hyperplastic reactive cell proliferation.	Iatrogenic KS mostly aneuploid pattern, classic KS diploid pattern on flow cytometry.

♂: Male, ♀: female, aCGH: array comparative genomic hybridization, AIDS: AIDS associated Kaposi Sarcoma, Ca: Cases Ct: Controls, FISH: Interphase fluorescence in situ hybridization, Mol.An: Molecular analysis, HP: Histopathology, IHCS: Immunohistochemical staining, and SG: Staging. When data not reported in the table means the original article did not provide them.

Descriptions of clinical and morphological (histopathological) descriptions, such as plaque, nodular and patch stage, were frequently incomplete but still showed an important diversity of case combinations when studying the different outcomes, as follows:Clinical and morphological description of clonality studies: 34 (74%) of the 46 cases were HIV associated, 12 (16%) classic KS forms of all 17 (37%) studied nodular lesions, 3 (7%) were plaque lesion samples, and the 26 (57%) remaining cases were from 2 studies [25,27], which did not specify morphological characteristics of the lesions.Clinical and morphological description of mutation studies: in total, 84 (29%) HIV associated cases were included, 146 (52%) classic KS and 2 (1%) cases of iatrogenic KS. For 49 (17%) cases, no clinical information was available since Nicolaides et al. [28], comprising 31 cases, did not describe their clinical characteristics and Feller et al. [32] described only 6 of 18 included cases. All together reported that 20 (7%) were nodular lesions, 27 (10%) plaque lesions and 17 (6%) patch lesions, but these data were missing for 217 (77%) cases since 7 [21,22,29,30,33,34,35] of the 10 included studies did not specify the morphological characteristics of the lesions. Two studies reported separately for HHV8 positivity, Tornesello et al. [21] with all 67 cases reported as positive and Cordiali-Fei et al. [33] detecting in all the three cases of the studied family high titres of anti-HHV-8 (type A virus) antibodies.Clinical and morphological description of chromosomal aberrations studies: 58 (27%) of the 214 cases were HIV associated, 147 (69%) classic and 9 (4%) iatrogenic KS presentations. Only 3 studies [36,37,39] reported on morphological stages of these cases summing 3 (2%) plaque, 37 (17%) nodular, 23 (11%) lesion samples and 151 (70%) cases where it was not reported.Outcomes assessing the clonal nature of KS provided by the 20 included studies (Table 2, Table 3 and Table 4) resulted very heterogeneous (Figure 2). For a more comprehensive synthesis of the results, we decided to report retrieved outcomes grouped into the mentioned three categories of studied genetic alterations. However, differences in study aims, outcome definitions, applied methods and reported outcome measures limited seriously our possibility to pool and/or compare the published data. This heterogeneity in reported data was also the reason for ruling out a quantitative synthesis of the retrieved results, and the performance of a meta-analysis was excluded.Determining clonality: outcomes on clonality in KS samples were mostly determined by monoclonal patterns of gene inactivation or methylation and obtained mixed results. Four case series [24,25,26,27] obtained results suggesting that KS is a clonal neoplasm, while one study obtained results compatible with the description of a polyclonal cell [23]. All studies presented a high risk of bias as assessed by the adapted Quadas-2 tool due to bias inherent to their study design and insufficient reporting.Detecting mutations: Ten studies aimed to detect mutations in KS lesions, focusing three on p53 mutations [21,34,35] and seven on other different single mutations (IL-6, c-myc, LEC and BEC gene signatures, FGF3, HLA, FGF4 and KRAS). Each study applied different laboratory technics, determined different outcomes and reported diverse findings. Of these ten studies, three [21,29,32] reported negative results for the investigated mutation, and the other seven obtained outcomes suggested a possible relationship. Of the three studies [21,34,35] focusing on a possible role of p53 mutation in the KS oncogenesis, two [29,35] obtained results that point to a possible implication, and one case control study [21] with a large sample and well-performed statistical analysis failed in obtaining evidence of a possible association. These studies presented a high risk of bias, including the prospective case series [22] and the case control study [21] due to difficulties in assessing the internal and external validity of the studies based on the reported data.Demonstrating chromosomal aberrations: Finally, of the five selected studies [36,37,38,39,40] that investigated chromosomal aberrations, two studies [36,38] reported diploidy Reizis et al. [38] (a type of pattern for iatrogenic forms of KS) and aneuploid patterns for the classic form of KS. Another study [40] detected low levels of DNA aneuploidy, and the other two [37,41] reported results compatible with a hyperplastic reactive cell proliferation. Again, a high risk of bias was detected for all included studies due to limitations inherent to the study design and through a lack of detailed reporting of methods. Table 2, Table 3 and Table 4 summarize our findings.

## 4. Discussion

We performed a systematic review of the peer-reviewed and published scientific literature on genetic alterations assessing the clonal nature of KS. The limited number of retrieved studies and the low-level of evidence of the retrieved studies [42,43], together with the low number of recent publications investigating this topic suggests that this potentially controversial topic has not been studied in depth from a broad, multidisciplinary perspective. Scientific interest in recent years has focused mainly on describing the viral oncogenic role through mechanistic studies [44,45,46,47,48] and relatively little attention has been paid to the analysis of factors influencing clonal growth in KS. However, the identification of such underlying mechanisms and subsequent definitions could lead to improvements in the clinical management of KS patients.

That said, there are several aspects of the pathogenesis of KS that are currently under investigation to determine the relevance of latency-associated nuclear antigen (LANA) and other HHV8 proteins in subverting normal endothelial cell biology to induce proliferation [49,50] and to describe the miRNA-34A effect [51]. Evidence from basic research proposing that HHV8 is an integrating oncovirus that causes amplification and activation of oncogenes needs to be further investigated by the sequencing of early and advanced lesions considering the latent state in which HHV8 has been reported to exist [52] as the clonality of HHV8 integration is sometimes questioned. While many studies have been published reporting genetic alterations in KS samples, few have analysed whether these are driver mutations that might be involved in oncogenesis [53,54]. 

We retrieved only two studies with an appropriate study design for the investigation of such associations; however, both focused on epidemiological aspects of the disease and did not provide effect measures analysing the relationship of a mutation or clonality with KS. Nicolaides et al. [28] showed *KRAS* mutation of amplification in substantial numbers of advanced cases in which codons 13, 15, 16, 18 and 31 were implicated. A small number of patients with the TP53 mutation have been found and may be associated with progression [34,35]. TP53 immunohistochemistry has been used, and expression appears to increase with progression [55]. The ability of NGS to examine intra-tumoral heterogeneity, mutation load, copy number variation and temporal heterogeneity makes this a major gap in the current knowledge of KS.

Clonality of the virus can be shown [56,57] but is not direct evidence of clonality of the host cells. Indeed, this may simply represent clonal restriction following polyclonal infection [23]. However, in a large study of 98 patients, Duprez et al. [58] used an HHV-8-fused terminal repeat (TR) to follow up an earlier study [59] to show that most lesions were oligoclonal and concluded that KS was a reactive process in most patients, although many samples were not informative [58]. 

They found that differences ‘strongly suggest that disseminated lesions represent multiple distinct primary expansions of HHV-8-infected spindle cells originating from different infectious events rather than metastatic proliferations’. It is known that EBV-induced lymphoproliferative disease starts as an oligoclonal proliferation and can lead to monoclonal diffuse large B-cell lymphoma with the adoption of additional genetic mutations; however, the same has not been shown for KS—although it remains a possibility.

The chronic and slow evolution of classic-KS lesions, the partial reversibility of iatrogenic-KS lesions after diminution of immunosuppressive regimens and the complete epidemic-KS regression after anti-retroviral therapy seem more consistent with reactive proliferation [23,26,38,40]. Although there are occasional reports of spontaneous regression of individual lesions (and rarely of all disease) [60]. Dependence on cytokines, such as basic fibroblast growth factor, oncostatin M and interleukin 6, has been shown for the growth of Kaposi’s sarcoma-derived cells in vitro [31]. However, a commonly used cell line (KS Y-1) derived from KS lesions does produce tumours in nude mice [61].

Detailed, and ideally longitudinal, NGS studies with examination of multiples of a range of early (cutaneous patch stage) and late (visceral) lesions in individual patients could provide evidence of clonal selection and progression. Such an approach would evaluate for neoplastic transformation of early lesions.

## 5. Conclusions

In conclusion, insufficient scientific evidence exists to describe the nature of KS. Current knowledge is insufficient to determine whether KS is a clonal neoplasm, or simply an aggressive reactive virus-driven lesion. Although evidence from mechanistic studies that points towards a reactive virus-driven process that gives rise to a clonal neoplasm is mounting, direct evidence for the pathogenic pathway would allow an evidence-based assessment of KS neoplastic status with all the implications that this may have for clinical practice and patient care. Its current neoplastic classification as a sarcoma is questionable, and regarding it as a ‘Kaposi tumour’ may be more accurate.

## Figures and Tables

**Figure 1 cancers-14-01201-f001:**
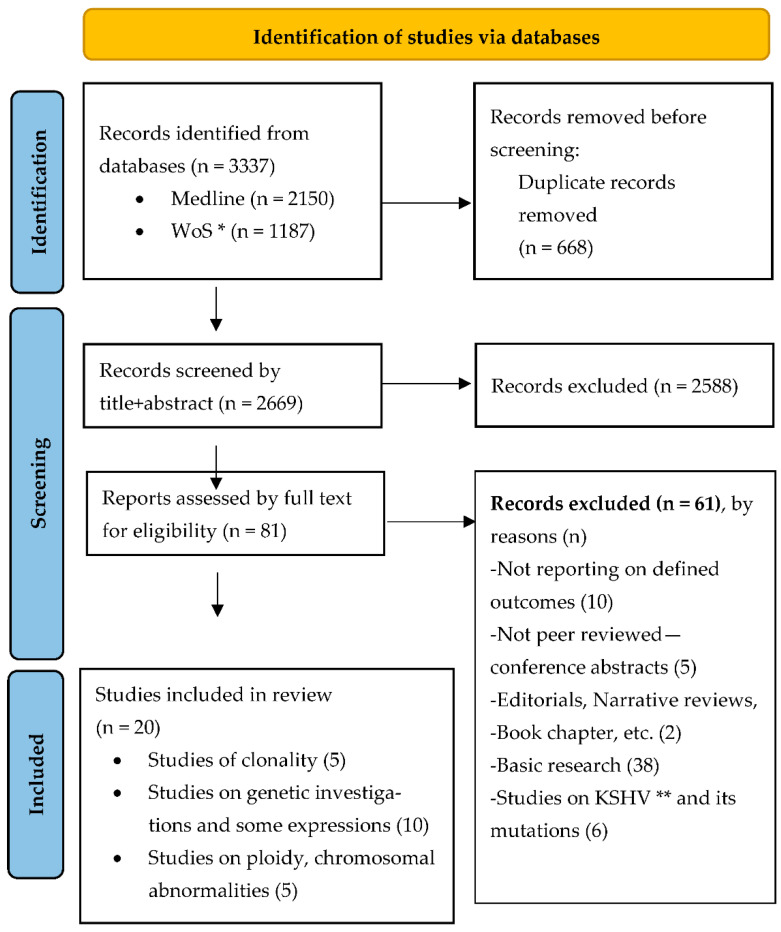
PRISMA 2020 flow diagram [19] for systematic review on clonality and mutation in Kaposi sarcoma. * Web of Science, ** Kaposi Sarcoma related Herpes Virus research. Copyright statement: this PRISMA diagram contains public sector information licensed under the Open Government Licence v3.0. Adapted From: Moher D, Liberati A, Tetzlaff J, Altman DG, PLOS Medicine (OPEN ACCESS) Page MJ, McKenzie JE, Bossuyt PM, Boutron I, Hoffmann TC, Mulrow CD, et al. The PRISMA 2020 statement: an updated guideline for reporting systematic reviews. PLOS Medicine 2021;18(3):e1003583. doi: 10.1371/journal.pmed.1003583.

**Figure 2 cancers-14-01201-f002:**
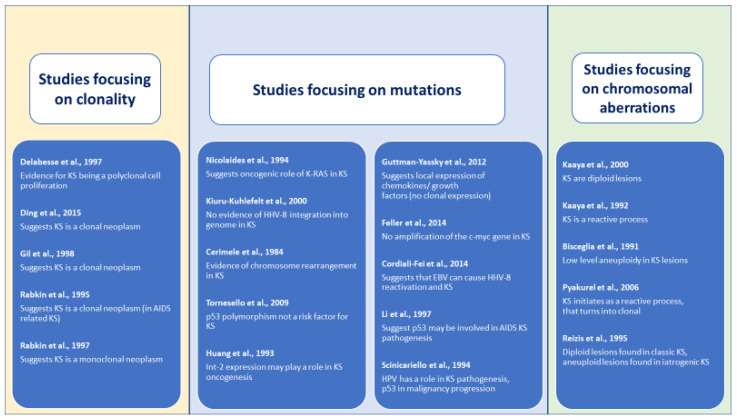
Outcomes of included studies assessing the neoplastic nature of Kaposi sarcoma.

**Table 1 cancers-14-01201-t001:** Search terms and strategies in MEDLINE and Web of Science.

Search term	Keywords	MeSH	Records
Concept 1			
Kaposi sarcoma	Kaposi * AND sarcoma *	Sarcoma, Kaposi	1988
Concept 2			
Clonality	Clonal *	Clonal Evolution	2012
Mutation	Monoclonal *	Mutation	1964
	Oligoclonal *	Polymorphism, Genetic	2005 (1968)
	Polyclonal *	Clone Cells	1968 (1964)
	Mutant *	Cell Proliferation	2005
	Reactive *		

Studies were included if they met the predetermined inclusion criteria: (a) Laboratory studies of KS biopsies, (b) peer-reviewed publications published in English, French, German or Spanish and (c) reporting original data on clonality or somatic mutations. Truncation symbol * was used to search for the root of the term only.

## Data Availability

All data generated or analysed during this study are included in this published article and its Appendix A.

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
