# Peer review of "Clonality, Mutation and Kaposi Sarcoma: A Systematic Review"

_cancers, 2022, doi:10.3390/cancers14051201_

Round 1
Reviewer 1 Report
In this manuscript, the authors have systematically reviewed current literature to determine whether KS is a clonal neoplasm (monoclonal) or aggressive reactive virus-driven lesion. A total of 20 publications were included to assess the neoplastic nature of Kaposi Sarcoma. The previous studies had considerable heterogeneity of data with a high risk of bias and therefore, authors reported existing evidence is insufficient to describe the nature of KS. The authors have comprehensively summarized and critically evaluated the merits of previous publications, and it will guide further investigation to resolve the issue.
MINOR REVISIONS:
Line 310 ' Author mentioned that the '….HHV-8 is an integrating oncovirus….' However, as cited in reference (52; Mui et al., 2017) and reported by many others, the HHV-8 in the latent state exists as an episome where LANA protein tethers circular genome to host chromatin. The viral genome is not integrative into the host genome.
Line 325: '…..and look at viral integration….' virus maintained as an episome.
Author Response
General comments reviewer 1:
In this manuscript, the authors have systematically reviewed current literature to
determine whether KS is a clonal neoplasm (monoclonal) or aggressive reactive
virus-driven lesion. A total of 20 publications were included to assess the neoplastic nature of Kaposi Sarcoma. The previous studies had considerable heterogeneity of data with a high risk of bias and therefore, authors reported existing evidence is insufficient to describe the nature of KS. The authors have comprehensively summarized and critically evaluated the merits of previous publications, and it will guide further investigation to resolve the issue.
MINOR REVISIONS:
Line 310 ' Author mentioned that the '….HHV-8 is an integrating oncovirus….'
However, as cited in reference (52; Mui et al., 2017) and reported by many others, the HHV-8 in the latent state exists as an episome where LANA protein tethers circular genome to host chromatin. The viral genome is not integrative into the host genome.
Line 325: '…..and look at viral integration….' virus maintained as an episome.
Response: Thank you for pointing this out. We agree with the reviewer that this isn´t coherent with the referenced literature and we therefore changed the wording in both lines. New wording reads as follows:
Discussion, Page 3 lines 310-313: Evidence from basic research proposing that HHV8 is an integrating oncovirus that causes amplification and activation of oncogenes needs to be further investigated by sequencing of early and advanced lesions considering the latent state in which HHV8 has been reported to exist (52), as the clonality of HHV8 integration is sometimes questioned.
Discussion, Page 3 lines 325-326: To date, there have been few published studies using NGS in KS. The ability of NGS to examine intra-tumoral heterogeneity, mutation load, copy number variation and temporal heterogeneity makes this a major gap in the current knowledge of KS.
Reviewer 2 Report
This review aims at collecting sufficient data to determine whether Kaposi sarcoma is a true neoplasm or rather a reaction to viral infection. Using a strict strategy to identify representative papers 20 out of 3337 studies were included. However no consensus on whether or not Kaposi sarcoma as a clonal neoplasm has emerged from this analysis. According to the authors this can be attributed to a lack of multidisciplinary approach.
The introduction is clearly written and quite informative without being too lengthy
The supplementary materials are unfortunately missing. I have asked the editor, whether I have overlooked the file, but it was confirmed that the supplement was not included in the submission.
Perhaps the standardized data extraction form can be included in the supplement to gain more insight into the selection of the papers
The tables 2-4 are quite complicated and not easy to read. Since all studies suffer from a high risk bias, this line in the table can be omitted. However fig 2 nicely summarizes the data.
The discussion lacks a recommendation on further investigation to resolve the lack of proof for Kaposi sarcoma as a true neoplasm. A possible solution could by next generation sequencing. The authors indeed refer to this technology. On line 323 they mention that few published studies are available, however this is not supported by references to these few studies.
Author Response
Comments reviewer 2:
This review aims at collecting sufficient data to determine whether Kaposi sarcoma is a true neoplasm or rather a reaction to viral infection. Using a strict strategy to identify representative papers 20 out of 3337 studies were included. However no consensus on whether or not Kaposi sarcoma as a clonal neoplasm has emerged from this analysis. According to the authors this can be attributed to a lack of multidisciplinary approach.
Response: Thank you to the reviewer for recognizing the relevance of this topic and helping us to improve the reporting and comprehension of of our work. All suggestions have been addressed as follows:
Point 1: The introduction is clearly written and quite informative without being too lengthy. The supplementary materials are unfortunately missing. I have asked the editor, whether I have overlooked the file, but it was confirmed that the supplement was not included in the submission. Perhaps the standardized data extraction form can be included in the supplement to gain more insight into the selection of the papers.
Supplementary materials have been uploaded.
Point 2: The tables 2-4 are quite complicated and not easy to read. Since all studies suffer from a high risk bias, this line in the table can be omitted. However fig 2 nicely summarizes the data.
Results, Tables 2-4: Line showing risk of bias evaluation results has been eliminated from all tables.
Point 3: The discussion lacks a recommendation on further investigation to resolve the lack of proof for Kaposi sarcoma as a true neoplasm. A possible solution could by next generation sequencing. The authors indeed refer to this technology. On line 323 they mention that few published studies are available, however this is not supported by references to these few studies.
The following statement has been added to the discussion (Discussion, lines 347-350): Detailed, and ideally longitudinal, NGS studies with examination of multiples of a range of early (cutaneous patch stage) and late (visceral) lesions in individual patients could provide evidence of clonal selection and progression. Such an approach would evaluate for neoplastic transformation of early lesions.